# Landau Diamagnetism and de Haas–van Alphen Oscillations, Formed in Single Crystals of Y_3_Fe_5_O_12_, in Local Nanodimensional-Sized 2D Phase Separation Regions, Located inside Layered Domain Walls at Room Temperature and T = 77 K

**DOI:** 10.3390/nano13142147

**Published:** 2023-07-24

**Authors:** Boris Khannanov, Evgeny Golovenchits, Mikhail Shcheglov, Viktoriya Sanina

**Affiliations:** Ioffe Institute, 194021 St. Petersburg, Russia; boris.khannanov@gmail.com (B.K.); e.golovenchits@mail.ioffe.ru (E.G.); m.scheglov@mail.ioffe.ru (M.S.)

**Keywords:** Landau diamagnetism, de Haas–van Alphen oscillations, nano dimensional 2D phase separation regions, domain walls, superlattices

## Abstract

This paper presents results of the magnetic dynamics study (the microwave power absorptions at the fixed frequencies during magnetic field sweeping) in samples of Y_3_Fe_5_O_12_ single crystals in the form of plates and spheres of various sizes, at frequencies exceeding 30 GHz, in magnetic fields up to 18 kOe, at room temperature, and T = 77 K. It was found that in this case, the inhomogeneity’s of the magnetic state manifested itself in the Y_3_Fe_5_O_12_ samples as 2D local phase separation regions. Such 2D phase separation regions formed inside layered domain walls representing superlattices with sizes of 700–900 Å. Depending on the shape and size of the studied plates and spheres, Landau diamagnetism or de Haas–van Alphen oscillations were observed in the 2D phase separation regions at room temperature and T = 77 K.

## 1. Introduction

Single crystals of yttrium iron garnet Y_3_Fe_5_O_12_ (YFeG), with cubic symmetry O_h_^10^, contain Fe^3+^ ions occupying two different positions: three ions in oxygen tetrahedrons and two ions in oxygen octahedrons. Nonmagnetic Y^3+^ ions are located in oxygen dodecahedrons. The spins of Fe^3+^ ions in different positions in the crystal have opposite orientations, forming a resulting nonzero magnetic moment (the ferrimagnetic state with Neel temperature T_N_ = 560 K) [1]. When studying the ferromagnetic resonance (FMR) of YFeG single crystals, which have the form of spheres of a rather small diameter, at low microwave frequencies in magnetic fields of the order of 3–4 kOe, it was found that YFeG behaves as a homogeneous, single-domain ferromagnetic with a narrow FMR line, which indicates low magnetic and dielectric losses [1]. For this reason, YFeG is the main material for applications in controlled microwave devices, operating at room temperature at sufficiently low microwave frequencies [1].

As noted above, YFeG is a ferrimagnet. The presence of several magnetic sublattices in ferrimagnets leads to the existence of several branches of ferrimagnetic resonances. The low-frequency branch of the ferrimagnetic resonance corresponds to the excitation of the magnetization vector precession in the effective field H_eff_, which is the sum of the external magnetic field, the anisotropy field, and the demagnetizing field. The precession of the sublattices magnetization vectors occurs in such a way that the antiparallelism of the magnetization vectors of sublattices is not violated. In this case, the ferrimagnetic resonance frequency ω = γ_eff_ H_eff_. This type of ferrimagnetic resonance is no different from FMR. The specificity of ferrimagnetic resonance appears only in a change in the magnetomechanical ratio γ_eff_. In the simplest case of ferrimagnet with two sublattices, having magnetizations M1 and M2, γ_eff_ = (M1 − M2)/(M1/γ_1_ − M2/γ_2_). Here, γ_1_ and γ_2_ are magnetomechanical ratios for sublattices [1].

The high-frequency branches of the ferrimagnetic resonance correspond to such types of the magnetization vectors of the sublattices precession, in which their antiparallelism is violated. These branches are characterized as exchange resonances and they are located in the infrared range of the spectrum [1].

In the presented work, the task was to study the magnetic dynamics (microwave power absorption during magnetic field growth) in YFeG single crystals of various shapes and sizes at relatively high frequencies above 30 GHz, in magnetic fields up to 18 kOe, at room temperature. It was found that in this case, the inhomogeneity of the magnetic state manifested itself in the YFeG samples due to the formation of nanosized local 2D regions of phase separation.

The aims of this work were to study the properties and mechanisms of formation of such 2D phase separation regions, as well as their influence on the magnetic dynamics under the above conditions.

Previously, it was found in [2] that at T = 5 K, at a frequency of 32.8 GHz, in magnetic fields from 8 to 16 kOe, the 2D phase separation regions that formed in YFeG and in EuMn_2_O_5_ (EMO) and Eu_0.8_Ce_0.2_Mn_2_O_5_ (ECMO) multiferroics turned out to be similar (Figure 1a and Figure 1b, respectively), although the nature of their formation was different.

An important circumstance responsible for the formation of phase separation regions in EMO and ECMO multiferroics was the presence in these crystals of the same number of 3d ions of different valence Mn^3+^–Mn^4+^ (charge ordering) [3,4,5]. The observation in YFeG of the FMR lines sets similar to the set of such lines in EMO and ECMO multiferroics (Figure 1a,b) [2] indicates that there are local regions in YFeG crystals, the properties of which are similar to those of the phase separation regions in EMO and ECMO [3,4,5]. In this case, as in EMO and ECMO, the phase separation regions in YFeG should present superlattices of ferromagnetic layers with different ratios of 3d Fe ions with different valences and charge carriers of different signs [3,4,5]. Note that, as shown in [3], in the EMO and ECMO multiferroics, the phase separation regions arose inside the magnetic domains of the matrix, which initially contained Mn ions of different valences (Mn^3+^, Mn^4+^). In this case, they were concentrated near the lattice distortions inside the domain walls [5], whereas in YFeG, phase separation regions appear directly in the nanosized domain walls (superlattices), located between the homogeneous magnetic domains of the matrix [3,6].

It was shown in [7,8] that in the domain walls of magnetic crystals, located between bulk domains with oppositely oriented magnetic moments, inhomogeneous states arise with local mechanical, magnetic, and electrical (flaxy-magnetoelectric (FME)) distortions. This effect should be take place in all magnetic crystals, including YFeG. With sufficiently strong FME distortions, the symmetry group in the domain walls of YFeG decreases to a noncentral one. In this case, an internal electric field arises in the domain walls, which increases their energy. Such a state lowers the energy due to the flow of free carriers from homogeneous domains to domain walls. It changes the charge states of the ions inside the domain walls (i.e., ions with variable valences appear inside the walls). This makes it possible to form the 2D phase separation regions in YFeG domain walls (superlattices) similar to those that appeared in EMO and ECMO multiferroics in the original domains (Figure 1a,b). X-ray studies of EMO and ECMO in [6] show that the sizes of these superlattices were 900 Å and 700 Å, respectively.

As follows from the above, in order for the phase separation regions to appear in the YFeG domain walls, similar to those that appear in EMO and ECMO multiferroics, the mobile electrons, changing the states of the domain walls, must be present in the uniform domains of the YFeG matrix. Below, we consider the cause of such electrons’ appearance in the YFeG matrix and their influence on the formation of the phase separation regions properties. We will also discuss the reason for the appearance of the similar phase separation regions in YFeG and in EMO and ECMO in a wide range of magnetic fields of 8–16 kOe, at T = 5K, at the frequency of 32.8 GHz (Figure 1a,b).

In this work, when studying the magnetic dynamics of phase separation regions in YFeG, it was found that at sufficiently high frequencies and, accordingly, in sufficiently strong magnetic fields, at room temperature, depending on the shape and size of the studied plates and spheres of YFeG, either Landau diamagnetism [9,10] or de Haas–van Alphen oscillations [11,12] were observed. These states were formed in the local nanosized 2D phase separation regions located inside layered domain walls (superlattices) with sizes of 700–900 Å.

## 2. The Mechanism of Phase Separation Regions Formation in Y_3_Fe_5_O_12_ Plates and Their Properties at High Frequencies, Room Temperature, and T = 77 K

As noted above, in the ferrimagnet YFeG 3d ions, Fe^3+^ occupies two different positions in the crystal: three ions in oxygen tetrahedrons and two ions in oxygen octahedrons. In this case, the spin orientations of Fe^3+^ ions in tetrahedrons and octahedrons are opposite. As a result, the YFeG matrix between pairs of cations (Fe^3+^ ions) always contains ions of oxygen anions (O^2−^) and the direct exchange interaction between Fe^3+^ ions is significantly weakened. In YFeG the indirect exchange [13,14] between pairs of Fe^3+^ ions through oxygen anion O^2−^ ions dominates due to hybridization of Fe^3+^–O^2−^–Fe^3+^ ions states. This differs significantly from the situation in EMO and ECMO multiferroics, in which direct double exchange is possible between neighboring pairs of ferromagnetic oriented manganese ions of different valence Mn^3+^–Mn^4+^ (in this case, the electrons that recharge these ions are mobile) [3,4,5,6].

The observed similarity of phase separation regions in YFeG and in EMO and ECMO at low temperature (T = 5K) in strong magnetic fields (Figure 1a and Figure 1b, respectively) indicates the need for the existence of charge ordering and mobile charge carriers in the phase separation regions in YFeG also. However, direct double exchange of mobile electrons in YFeG between pairs of Fe^3+^ ions in Fe^3+^–O^2−^–Fe^3+^ triplets in a weak magnetic field is not possible due to the opposite orientation of the Fe^3+^ ions spins in neighboring octahedrons and tetrahedrons. But when sufficiently strong magnetic fields are applied, it can be expected that in triplets of neighboring Fe^3+^–O^2−^–Fe^3+^ ions, the spins of Fe^3+^ ions and orbital electrons of O^2−^ ions can be oriented parallel to each other (along the magnetic field). As a result, in these triplets of ions, the appearance of the double exchange with the transfer of electron spins of the oxygen ion between ferromagnetic pairs of Fe^3+^ ions becomes possible. This leads to the formation of pairs of Jahn–Teller ions Fe^2+^–Fe^4+^. Such a process is energetically favorable as the energy decreases when the pits appear, formed by the Jahn–Teller effect. Since the entire YFeG crystal consists of similar triplets of Fe^3+^–O^2−^–Fe^3+^ ions, as the magnetic field increases, the process described above takes place in the ever-increasing volume of the YFeG crystal. In this volume with reduced energy, mobile electrons accumulate and the state with localized electrons switches to the state with mobile electrons. In this case, the abovementioned process of the charge ordering formation of Fe ions in YFeG in a wide range of strong magnetic fields of 8–16 kOe at T = 5 K turns out to be advantageous, which ensures the appearance of a sufficiently high concentration of free carriers, leading to the formation of phase separation regions, shown in Figure 1a. As noted above, for the formation of the state in YFeG with five intense lines (Figure 1a), a rather high concentration of carriers is required in the matrix domains, whose overflow into the domain walls forms such regions of phase separation with five intense lines (Figure 1a).

The process described above for YFeG corresponds to the situation considered in [15], where it was shown that the Mott transition in orbital degenerate systems can occur (and often occurs) not in the standard sequence “Mott insulator—weakly correlated metal”, but through a new intermediate phase with charge (rather than orbital) ordering. In this case, the removal of orbital degeneracy and the appearance of charge ordering is an alternative to the Jahn–Teller distortion. This occurs when switching from a state with localized electrons to a state with mobile electrons takes place. In this case, it turns out to be energetically favorable to connect the intra-atomic Hund’s rule, which orients the spins of mobile electrons parallel, thus forming the charge ordering of Fe ions of different valence, instead of the local Coulomb repulsion, which prevents the formation of such ordering. In [15], this was shown not only theoretically, but also in the example of experimental situations in nickelates [16,17,18,19,20,21] and in compounds with iron ions [22,23].

We believe that the situation described in [15,22,23] is also realized in the domain walls of the YFeG plate (with dimensions of 4 × 3.5 × 1.4 mm), in which charge orderings of no Jahn–Teller pairs ions [Fe^5+^(3d_1_)–Fe^1+^(3d_5_)] arise in sufficiently strong magnetic fields. As a result, a new state appears in the domain walls of such a plate in YFeG, characterized by the presence of double exchange, free carriers, and charge ordering, which ensures the appearance of phase separation regions in YFeG, similar to the regions in EMO and ECMO (Figure 1a,b). The possibility of direct conversion of pairs of Fe^3+^–Fe^3+^ ions into pairs of ions [Fe^5+^(3d_1_)–Fe^1+^(3d_5_)] in the domain walls at a sufficiently high concentration of carriers in the domain walls is not ruled out. Preliminary Jahn–Teller distortions of [Fe^2+^(3d_4_)–Fe^4+^(3d_2_)] ion pairs in domain walls contribute to an increase in the electron concentration in them. As a result, the 2D phase separation regions that emerge in YFeG in the domain walls are a set of 2D layers containing a different number of localized magnetic Fe ions of different valences and mobile charge carriers with different concentrations and signs (Figure 1a) [3].

Localized spins of Fe^3+^ ions in homogeneous YFeG domains in an applied magnetic field have a positive magnetic susceptibility, while in domain walls the magnetic susceptibility of the mobile electrons spins in 2D layers of phase separation can be negative. This negative magnetic susceptibility results from the rotation of mobile electrons in 2D layers around a magnetic field applied perpendicular to these layers. In this case, the orbital states of electrons are quantized in a magnetic field, forming the Landau diamagnetism [9,10].

In order for de Haas–van Alphen oscillations to be excited in 2D layers of phase separation regions in YFeG, the charge carriers in these layers must be free (i.e., the indirect exchange in triplets of (Fe^3+^–O^2−^–Fe^3+^) ions must be converted into a double exchange on a much larger scale containing many triplets of (Fe^3+^–O^2−^–Fe^3+^) ions). As noted above, this is possible under several conditions. Firstly, rather large regions with ferromagnetic ordering of localized spins of Fe ions and spins of mobile electrons should appear in 2D layers of phase separation regions in YFeG. This is possible when sufficiently strong magnetic fields are applied. In addition, the temperature must be high enough to increase the kinetic energy of the mobile electrons. When these two conditions are met, the mobile electrons can freely move within the plane of the 2D layer. However, displacement across the layers is difficult due to barriers at the layer boundaries, i.e., 2D electron gas appears in 2D layers of YFeG. This is the third necessary condition for the formation of de Haas–van Alphen oscillations [11,12].

It should be noted that at not very high temperatures (in our case, 300 K, compared with T_N_ = 560 K for YFeG), the thermal motion of atoms has little effect on the motion of electrons inside these atoms. As follows from the above, the properties and mobility of electrons in 2D phase separation regions in YFeG domain walls, which make it possible to form de Haas–van Alphen oscillations, require the application of sufficiently strong magnetic fields at room temperature.

## 3. Dependence of the Phase Separation Regions Properties on the Geometric Characteristics of the Plates

When studying the properties of 2D phase separation regions in YFeG in sufficiently strong magnetic fields and room temperature, it was found that the formed states of 2D phase separation regions turned out to be dependent on the sizes and shapes of the plates and spheres we studied. We studied three types of plates with dimensions 8.3 × 5.3 × 0.5 mm, 6.2 × 5 × 0.5 mm, and 4 × 3.5 × 0.4 mm, and spheres with diameters of 0.5 mm and 1.5 mm.

Y_3_Fe_5_O_12_ crystals were grown by the solution-in-melt method. Vertical furnace was used, the temperature regime of which is maintained with the required accuracy. In the zone of a moderate temperature gradient, conditions were created for the growth of large and homogeneous crystals. Y3Fe5O12 crystals had predominantly developed three [110], [111], and [001] faces. In these three directions, X-ray diffraction studies of Y_3_Fe_5_O_12_, presented below, were carried out. They confirmed the presence of nanosized superstructures in 2D phase separation regions. The X-ray study also showed that only the garnet structure is formed in yttrium garnet and there were no inclusions of regions with other structures. Spheres of various diameters were produced in the special machines with a conical abrasive groove, in which crystals rotated with compressed air.

The experimental conditions in our case are such that the studied elongated thin plates with dimensions of 8.3 × 5.3 × 0.5 mm and 6.2 × 5 × 0.5 mm, as well as a plate with dimensions of 4 × 3.5 × 0.4 mm, are oriented longitudinally along the sample holder (maximum dimension of plate along holder). The wave vector of the microwave wave (at frequency of 34.5 GHz) is perpendicular to these maximum planes of the plates, and the magnetic field is oriented along the average size of the plate. In this case, domains with opposite orientations of magnetic moments and domain walls between them are oriented along the magnetic field (i.e., along the average size of the plates). As a result, 2D layers of phase separation with 2D electron gas, formed in the domain walls of such plates, are located perpendicularly to the magnetic field. Such magnetic field leads to the rotation of mobile electrons in these layers. In this case, an additional magnetic field arises that has an opposite orientation with respect to the initial magnetic field, forming a diamagnetic moment (Landau diamagnetism) [9,10]. With an increase in magnetic field value, the diamagnetic response was enhanced, and with a sufficiently strong magnetic field, the state of phase separation regions in plates with dimensions of 6.2 × 5 × 0.5 mm and 8.3 × 5.3 × 0.5 mm changed abruptly (Figure 2 and Figure 3, respectively).

As can be seen from Figure 2 and Figure 3, in increasing magnetic fields, a rather weak response of the observed magnetic dynamics begins to appear only in a magnetic field, the magnitude of which approaches the field value corresponding to the FMR resonance ω = γH_0_, γ = 2.8 at given frequency (34.5 GHz). However, the uniform FMR is not observed in the elongated thin plates under study. A rather complex picture is observed due to the formation of an inhomogeneous state.

As noted above, 2D phase separation regions in YFeG formed in inhomogeneous, layered domain walls (superlattices). The presence of such regions is manifested in the appearance of rather weak signals with an increase in the magnetic field in Figure 2 and Figure 3 from the value of γH_0_ up to magnetic fields of 12.8 kOe and 13.75 kOe, respectively, for plates 6.2 × 5 × 0.5 mm (Figure 2) and 8.3 × 5.3 × 0.5 mm (Figure 3). These weak signals demonstrate that as the magnetic field increases, free electrons gradually accumulate in these plates, changing the charge states of the plates.

Finally, in magnetic fields of 12.8 kOe and 13.75 kOe, respectively, for plates 6.2 × 5 × 0.5 mm (Figure 2) and 8.3 × 5.3 × 0.5 (Figure 3), rather powerful narrow absorption jumps are observed, after which the response of 2D regions of phase separation states disappears. We note that the process of the phase separation regions formation in these plates is inertial and is gradually formed in the domain walls during successive cycles of magnetic field growth with an ever-decreasing field growth rate. The stable long-lived states, represented by the jumps in Figure 2 and Figure 3, formed at room temperature in the third cycle of magnetic field growth at a rate of 0.1 A/min. We attribute the observed absorption jumps to structural distortions caused by manifestations of Landau diamagnetism, which forms the equilibrium states of 2D phase separation regions.

As noted above, with an increase in the magnetic field in YFeG, a gradual change in the states of 2D phase separation regions in domain walls occurs. Initially, in lower magnetic fields, the recharging of pairs of Fe^3+^ ions by orbital electrons of oxygen ions (in triplets of Fe^3+^–O^2−^–Fe^3+^ ions) occurs and pairs of Jahn–Teller ions Fe^2+^–Fe^4+^ appear. This energetically favorable process lowers the energy due to Jahn–Teller lattice distortions near these pairs of ions. At the same time, the number of localized electrons increases in deeper Jahn–Teller wells, significantly enhancing the Landau diamagnetism, which reduces the initial number of free electrons in the 2D layers of phase separation regions. In this case, it is energetically more favorable not to form the final state of phase separation regions with charge ordering of Fe^5+^–Fe^1+^ ions (requiring the presence of excess free electrons), but, rather, a state with orbital ordering of Jahn–Teller Fe^4+^–Fe^2+^ ions, which leads to structural distortion of phase separation regions.

In a flat plate 6.2 × 5 × 0.5 mm, in a new equilibrium state formed after structural distortion of the phase separation regions (Figure 2), the conductivity and electric polarization were measured. The polarization was measured by the PUND method [24] in an electric field of 5 kV, applied perpendicular to the plane of the plate. It turned out that the measured conductivity and polarization were much lower than in the previously studied orthoferrites, orthochromites, and multiferroics RMn_2_O_5_ [2] and were practically difficult to measure. In this case, the small polarization in YFeG had a negative sign with respect to the direction of the electric field. This indicates a different nature of the conductivity and electric polarization formation in YFeG upon application of an electric field compared to the crystals mentioned above.

This situation can be understood, if we assume that in this plate with a low concentration of free charge carriers, it is energetically unfavorable to form a charge ordering of the Fe^5+^–Fe^1+^ ions, which requires a large number of such charge carriers. In this case, it is energetically more favorable to form the orbital ordering of the Fe^2+^–Fe^4+^ ions due to the Jann-Teller effect (Figure 2 and Figure 3). With the subsequent application of an electric field E = 5 kV to the sample in the orbital ordering state, inside the 2D phase separation regions there is a redistribution of free carriers in such a way that compensation of this applied electric field occurs and there is a change in the states of these 2D phase separation regions. At a sufficiently high concentration of free carriers in such regions, it is possible to change the sign of the low electric polarization observed by us.

A different picture of the 2D phase separation regions formation is observed when studying magnetic dynamics at a frequency of 34.5 GHz for a YFeG plate with dimensions of 4 × 3.5 × 0.4 mm in increasing magnetic fields at room temperature and T = 77 K. In Figure 4, in the first cycle of magnetic field growth at a rate of 0.1 A/min in this plate at room temperature, a number of absorption lines of 2D phase separation regions are observed, the most intense of which are marked with numbers 1–5. Such lines appear in a certain well formed by phase separation regions and distorting the background absorption value. At the same time, near the right jump of the background (in maximum magnetic fields starting from the magnetic field value of 12.27 kOe), rather weak oscillations are observed in the interval of magnetic fields of 12.27–12.53 kOe (Figure 4).

We note that a similar pattern with background jumps and a slope of the line connecting the centers of the absorption lines of the phase separation regions, observed by us earlier, under sufficiently strong optical pumping in a doped ECMO multiferroic containing 2D layers with charge carriers arose when EMO was doped with Ce^4+^ ions [6]. In [6], we associated such features of the absorption lines of the phase separation regions with the appearance of electric polarization in ECMO. In this case, during the subsequent cycling of the magnetic field, the electric polarization in ECMO gradually disappeared and the initial state of the phase separation regions was restored.

In contrast to ECMO, in the YFeG plate it turned out that over time, at room temperature, the state of the plate shown in Figure 4 began to change spontaneously without external influences, i.e., the state shown in Figure 4 is metastable. After 20 h of spontaneous relaxation, a new state appeared, shown in Figure 5, during its first measurement with a magnetic field growth with rate of 0.1 A/min.

As can be seen from Figure 5, background jumps were reduced. The background slope under the absorption lines also changed to the opposite direction. This indicates a spontaneous redistribution of charge carriers between the phase separation regions from the state in Figure 4 to the state shown in Figure 5. In this case, the nature of the absorption lines (as in Figure 4) changes near 12.27 kOe. In lower magnetic fields, broader absorption lines with weak distortions are observed. With an increase in the magnetic field from 12.27 kOe and up to a field of 12.53 kOe in Figure 5, oscillations also appear with a slightly higher intensity compared with Figure 4.

Finally, in the third cycle of the magnetic field sweep at a speed of 0.1 A/min (without a time pause after measuring the state in Figure 5), an equilibrium stable state of phase separation regions is formed with more intense absorption lines, shown in Figure 6. As can be seen from Figure 6, the background distortion disappeared completely, but as the magnetic field increased, a complex pattern of 2D regions formed, and the number of free electrons gradually increased. At first, in smaller fields, these charge carriers slightly distort the absorption lines in such 2D layers. In the same maximum fields (H > 12.27 kOe) and in the same range of magnetic fields 12.27–12.53 kOe, even more intense oscillations than in Figure 5 are observed. In this case, the intensity of the oscillations also decreases as the field of 12.53 kOe is approached. The YFeG state described above, shown in Figure 6, was reproduced at room temperature with new pulls of the magnetic field for a long time, i.e., it is an equilibrium stable state.

For the oscillations observed in Figure 4, Figure 5 and Figure 6, in magnetic fields in the range of 12.27–12.53 kOe at room temperature, we refer to the de Haas–van Alphen oscillations arising in 2D phase separation regions. Remember, that in YFeG, 2D phase separation regions appear directly in the nanosized domain walls (superlattices) located between the homogeneous magnetic domains of the matrix [3,6].

According to [11,12], de Haas–van Alphen oscillations arise when an electron beam, deflected by a magnetic field, crosses the Fermi levels in states formed in 2D layers with an increased level of charge carriers. This means that in YFeG plate with dimensions of 4 × 3.5 × 0.4 mm at room temperature, in all three cases of the magnetic field increasing with velocities of 0.1 A/min (Figure 4, Figure 5 and Figure 6), in the 2D layers in the range of magnetic fields of 12.27–12.53 kOe, the state close to a metallic one with Fermi levels is formed.

But at the same time, with the gradual formation of states in increasing magnetic fields (Figure 4, Figure 5 and Figure 6), the concentration of free electrons gradually increases in 2D layers (in domain walls with structural distortions, enhancing the metallization of domain walls). As noted above, the displacement of electrons across 2D layers from domain walls to homogeneous domains is difficult due to barriers at the boundaries of domain walls and domains. As a result, the 2D electron gas is formed in the domain walls.

The twisting of electrons in a transverse growing magnetic field in the domain walls leads to the intersection of electron fluxes of the Fermi levels and to the formation of de Haas–van Alphen oscillations [11,12].

As noted above, de Haas–van Alphen oscillations in the plate 4 × 3.5 × 0.4 mm (Figure 4, Figure 5 and Figure 6) begin to be excited in magnetic fields greater than the field corresponding to the condition for the occurrence of FMR ω = γH_0_ (H_0_ = 12.27 kOe). These oscillations exist only in the range of magnetic fields of 12.27–12.53 kOe of the existence of such a resonance. These values of magnetic fields at the frequency of microwave radiation used by us are within the width of the FMR line. We attribute the observation of de Haas–van Alphen oscillations in the range of magnetic fields of the existence of FMR to the fact that the spins of 3d Fe^3+^ ions, processing in a rather strong resonant magnetic field, and the spins of mobile electrons of oxygen ions, recharging these ions, are parallel to each other. This contributes to the enhancement of double exchange and the appearance of an increased concentration of mobile electrons in such 2D phase separation regions in which they are localized. As the magnetic field grows, a smaller number of free carriers participate in the further process of oscillation formation (minus carriers that are localized earlier), reducing the intensity of the observed oscillations. When the maximum magnetic field of the FMR existence is approached, the oscillation disappears. This is the reason for the decrease in the intensity of oscillations as it approaches the maximum field of existence of a ferromagnetic resonance in Figure 4, Figure 5 and Figure 6.

An important fact when choosing a frequency is the need to apply a sufficiently strong magnetic field that controls the states of 2D regions formed in YFeG in domain walls with flaxy-magneto-electric distortions. This also determines the choice of a sufficiently high frequency when studying such magnetic dynamics in sufficiently strong magnetic fields. As noted above, at low frequencies (9 GHz) in rather weak magnetic fields (3–4 kOe), a homogeneous single-domain state with narrow FMR lines is observed in a small YFeG sphere [1].

It is known [11,12] that the oscillation frequency, F, of de Haas–van Alphen oscillations in magnetic field units F = (π × c × ћ/e) × n = 2.06 × 10^−7^ × n is determined by the charge carrier concentration n in 2D layers. Here, c is the speed of light, e is the electron charge, and ћ is Planck’s constant. In our case, at room temperature (Figure 6), frequency F ≈ 10^4^ Oe, and charge carrier concentration n = 5 × 10^10^–10^11^ cm^−2^. The obtained value of the charge carrier concentration is at the lower limit of the possibility of the existence of de Haas–van Alphen oscillations [11,12]. For this reason, de Haas–van Alphen oscillations are observed in YFeG only inside 2D phase separation regions in maximal magnetic fields, in which the conditions for the excitation of such oscillations are realized. In 2D regions, observed in weaker magnetic fields, the conditions for the occurrence of de Haas–van Alphen oscillations are not met due to a decrease in the concentration of charge carriers.

As can be seen from Figure 5 and Figure 6, in a plate of 4 × 3.5 × 0.4 mm, in weak magnetic fields (lower than the field at which de Haas–van Alphen oscillations occur) in 2D phase separation regions, only low-intensity local distortions are observed, indicating structural distortions in these regions. We attribute this to the fact that as the applied magnetic field increases; the number of electrons in the 2D phase separation layers gradually increases. At their low concentration, local regions arise in which a small number of pairs of Fe^3+^–Fe^3+^ ions are recharged by electrons, forming pairs of Jann–Teller ions Fe^2+^–Fe^4+^, which contributes to the local structural distortion and further leakage of electrons into these local 2D regions. With a further increase in the field, the local 2D regions of phase separation are metallized and de Haas–van Alphen oscillations are formed in them.

As the temperature decreases (at T = 77 K), in the same plate, in the same maximum magnetic fields, de Haas–van Alphen oscillations with similar frequencies and carrier concentrations remain (Figure 7). However, as seen from Figure 7, their oscillation amplitudes are much smaller. As a result, it turned out that in YFeG, with decreasing temperature, the intensity of de Haas–van Alphen oscillations in the domain walls of YFeG decreases. The intensity of oscillations in the wider 2D regions, observed in magnetic fields below the value of the ferromagnetic resonance field at T = 77 K, also decrease, as compared with Figure 6.

As noted above, as the temperature in YFeG decreases, the kinetic energy of free electrons in the phase separation regions decreases at the same rate as its concentration. This leads to a decrease in their mobility in a magnetic field. As a result, at the same frequency, F, of de Haas–van Alphen oscillations in magnetic field units, the concentration of mobile electrons in the domain walls decreases, reducing the amplitude of their oscillations (Figure 7). This confirms the above statement that in order to excite more intense de Haas–van Alphen oscillations (Figure 6) it is necessary to apply sufficiently strong external magnetic fields at a sufficiently high (room) temperature. When these two conditions are met, mobile electrons can freely move within the plane of the 2D layer.

A comparison of Figure 2 and Figure 3 with Figure 4, Figure 5, Figure 6 and Figure 7 demonstrates the regions of phase separation in YFeG plates of various sizes and shapes at different temperatures, and shows that the properties of phase separation regions depend significantly on the shape and size of the samples. As indicated above, in thin elongated plates with dimensions of 6.2 × 5 × 0.5 mm and 8.3 × 5.3 × 0.5 mm, when placed in a magnetic field, de Haas–van Alphen oscillations were not formed, but a structural distortion was observed due to Landau diamagnetism, which forms a state of phase separation regions in such plates (Figure 2 and Figure 3), while in a higher-volume plate with dimensions of 4 × 3.5 × 0.4 mm, de Haas–van Alphen oscillations are formed (Figure 4, Figure 5, Figure 6 and Figure 7).

Thus, it turns out that the magnetic dynamics of YFeG crystals at room temperature significantly depend on the magnitude of the applied magnetic field, which changes the domain structure (homogeneous domains and domain walls with structural inhomogeneity). A change in the magnetic field mainly affects the change in the ratio of localized magnetic Fe ions with different valences and mobile electrons in 2D layers of phase separation regions. In this case, it is possible to form a stable equilibrium state of phase separation regions at room temperature in a plate with dimensions of 4 × 3.5 × 0.4 mm, which was preserved when the magnetic field was turned off (Figure 6).

## 4. X-ray Diffraction Study of the Y_3_Fe_5_O_12_ Garnet Crystal Structure

For a more in-depth analysis of the defects pattern in a Y_3_Fe_5_O_12_ sample of a cubic structure, we used reflections that characterize lattice distortions for the three main directions of the axes: [110], [111], and [001]. The measurements were carried out with copper radiation CuKλ1 (λ = 1.540562 Å). The sample in the form of a plate had a faceting along the natural growth faces with a base oriented along the (112) crystallographic planes and with well-developed lateral inclined faces along the (112) and (110) planes. All measurements were carried out in a single zone of one side face with planes of the (110) type.

In Figure 8, the results of measurements are presented in the form of reciprocal space maps (X-ray reciprocal space maps (RSMs)).

The diffraction pattern (Figure 8a) was obtained under the conditions of standard 880 symmetrical reflections from the face of the (110) planes. For two other cases (Figure 8b,c) with diffraction from (001) and (111) type planes from the same face, measurement schemes were used in conditions of skew geometry at sample tilt angles ψ in accordance with 00.12—(ψ = 45°) and 888—(ψ = 35.3°).

Structural features of YFeG observed were common to all three cases:(1)Significant distorting of the values of the lattice constant Δd.(2)High level of orientation ordering of the structure (half-width of rocking curves in the three-crystal diffraction mode: ω ~ 10′′).

This type of diffraction pattern is characteristic for nanoscale multilayer structures with periodic alternation of two layers differing in lattice parameter d. Moreover, between these layers there are areas with variable values Δd = d1 − d2, caused by deformations. Naturally, periodically alternating layers, differing in lattice parameter d, classify as domains. Regions with variable values of Δd = d1 − d2 caused by deformations are assigned to domain walls located between domains.

## 5. Study of the Spherical Samples Properties

We also studied the magnetic dynamics of a set of YFeG spheres with different diameters (0.5–0.6 mm and 1.5–1.6 mm), at room temperature, in strong magnetic fields at a frequency of 34.5 GHz. It turned out that in both spheres the 2D phase separation regions arosed. The properties of these regions depended on the orientation of the sphere with respect to the applied magnetic field and on the magnitude of this field. We present the results of a study of spheres at certain angles of their orientation, relative to the magnetic field, at which their clearest states were manifested.

It turned out that the properties of the 2D phase separation regions, arising in these two spheres of different diameters, were significantly different.

As shown in Figure 9, in the YFeG sphere with the minimum diameter of 0.5–0.6 mm, for a certain angle relative to the magnetic field, a set of five absorption lines, characterizing the 2D phase separation regions, were observed.

As follows from the above, the presence of five lines of phase separation regions with a similar ratio of line intensities was observed both in YFeG and in EMO and ECMO multiferroics (Figure 1a,b) at T = 5 K, in sufficiently strong magnetic fields. At the same time, in YFeG they formed in domain walls, while in EMO and ECMO they formed in homogeneous domains, there was a charge ordering of Fe^5+^–Fe^1+^ ions [3].

As can be seen from Figure 9, for a ball with a minimum diameter of 0.5–0.6 mm for the selected high frequency, phase separation regions were observed at room temperature in the strong magnetic fields of 12.0–12.6 kOe. In this case, it should be assumed that the magnetic state of such a small ball is homogeneous (there are no domains and domain walls). The observed five narrow lines of phase separation regions (Figure 9) most likely formed by the influence of the redistribution of mobile electrons in triplets of neighboring Fe^3+^–O^2−^–Fe^3+^ ions in an applied magnetic field when indirect exchange in YFeG is taken into account. Note that the observed five lines of phase separation regions in this small ball are also in the range of magnetic fields in which FMR is observed at the chosen measurement frequency. This means that the spins of Fe^3+^ and O^2−^ ions are parallel in the triplets of neighboring Fe^3+^–O^2−^–Fe^3+^ ions, and the magnetic fields are high enough for mobile electrons to appear in a small ball. These electrons recharge the neighboring pairs of Fe^3+^ ions in such a way that charge ordering of Fe^5+^–Fe^1+^ ions formed inside the matrix of the ball (Figure 9). In this case, the observed five lines of phase separation regions have a rather high signal intensity, similar to that observed for EMO and ECMO in Figure 1b, in which a similar mechanism for the formation of phase separation regions was realized.

Another picture of the phase separation regions states at room temperature was observed in a sphere 1.5–1.6 mm in diameter, at a certain orientation angle relative to the magnetic field, in the third cycle of magnetic field growth at a rate of 0.1 A/min (Figure 10).

In such a sphere, at room temperature, in the same range of magnetic fields for the existence of FMR (12.53 kOe > H > 12.2 kOe), de Haas–van Alphen oscillations arose with close frequencies and, consequently, with a carrier density close to that which took place in a YFeG plate with dimensions 4 × 3.5 × 0.4 mm (Figure 6). In this case, the de Haas–van Alphen oscillation intensity is close to that observed for a plate with dimensions of 4 × 3.5 × 0.4 mm (Figure 6) and also decreases as the magnetic field approaches H = 12.53 kOe. In a magnetic field of H ≈ 13 kOe, outside the FMR, an intense maximum is observed, more similar to structural distortions in Figure 2 and Figure 3.

We attribute this to the fact that in stronger magnetic fields, beyond the limits of the existence of FMR and de Haas–van Alphen oscillations, the concentration of mobile electrons significantly decreases, and were localized during the formation of the state in fields below 12.53 kOe. In this case, a more probable process in high fields is the recharging by the remaining electrons of a low concentration of only neighboring pairs of Fe^3+^ ions in the triplets of Fe^3+^–O^2−^–Fe^3+^ ions, thus forming pairs of Jann–Teller ions Fe^2+^–Fe^4+^. This energetically favorable process lowers the energy due to Jann–Teller lattice distortions near these pairs of ions. In this case, structural distortions are formed, similar to those observed in Figure 2 and Figure 3. Thus, in a more voluminous sphere of larger diameter, with increasing magnetic field, de Haas–van Alphen oscillations firstly form in the range of FMR magnetic fields, which absorb most of the mobile electrons. In stronger fields, a small number of mobile electrons during the charge exchange of Fe^3+^ ions form local Jahn–Teller structural distortions, which disappear with a further increase in the magnetic field.

## 6. Conclusions

As a result of the magnetic dynamics study in YFeG single crystals in the form of plates and spheres of various sizes, it was found that the 2D local phase separation regions formed in the domain walls, representing super lattices with sizes 700–900 Å, significantly affect the magnetic dynamics. The magnetic dynamics studies were produced at frequencies exceeding 30 GHz in magnetic fields up to 18 kOe at room temperature and T = 77 K. It was found that, depending on shape and size of the studied plates and spheres of YFeG samples, Landau diamagnetism or de Haas–van Alphen oscillations were observed in the 2D local phase separation regions.

## Figures and Tables

**Figure 1 nanomaterials-13-02147-f001:**
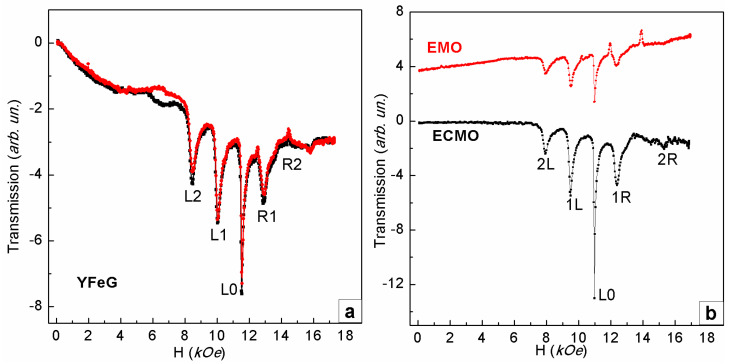
(**a**) Phase separation regions in YFeG plate with dimensions of 4 × 3.5 × 1.4 mm at T = 5 K, at frequency F = 32.8 GHz, H||a, k||c. (**b**) Phase separation regions in EuMn_2_O_5_ and Eu_0.8_Ce_0.2_Mn_2_O_5_, at T = 5 K, at F = 32.8 GHz, H||a, k||c [2].

**Figure 2 nanomaterials-13-02147-f002:**
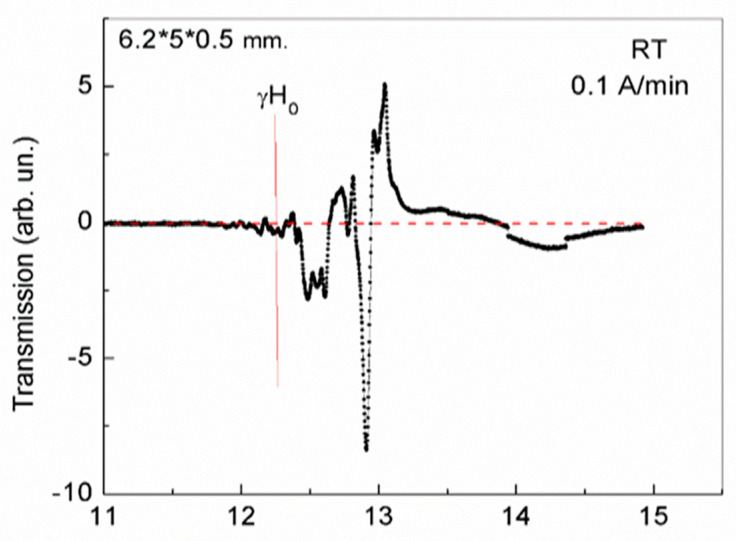
Magnetic dynamics at a frequency of 34.5 GHz for 2D phase separation regions in YFeG plate with dimensions of 6.2 × 5 × 0.5 mm at room temperature (RT) in an increasing magnetic field at a rate of 0.1 A/min.

**Figure 3 nanomaterials-13-02147-f003:**
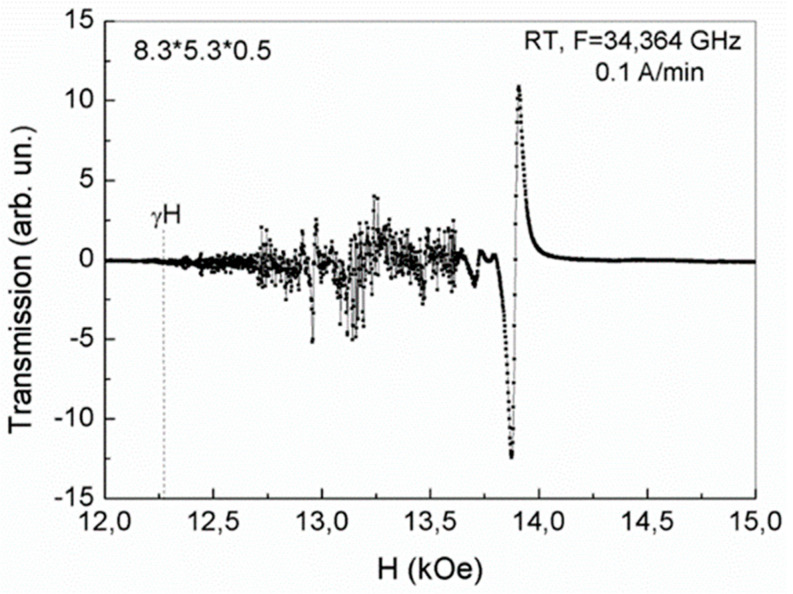
Magnetic dynamics of 2D phase separation regions at a frequency of 34.64 GHz in YFeG plate with dimensions of 8.3 × 5.3 × 0.5 mm at room temperature (RT) in an increasing magnetic field at a rate of 0.1 A/min.

**Figure 4 nanomaterials-13-02147-f004:**
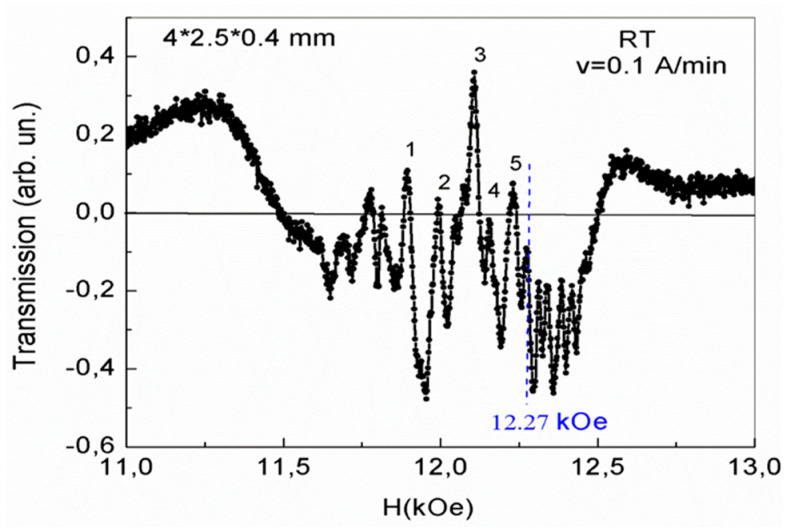
Formation of phase separation regions in a plate with dimensions 4 × 3.5 × 0.4 mm at room temperature in the first cycle of magnetic field growth at a rate of 0.1 A/min.

**Figure 5 nanomaterials-13-02147-f005:**
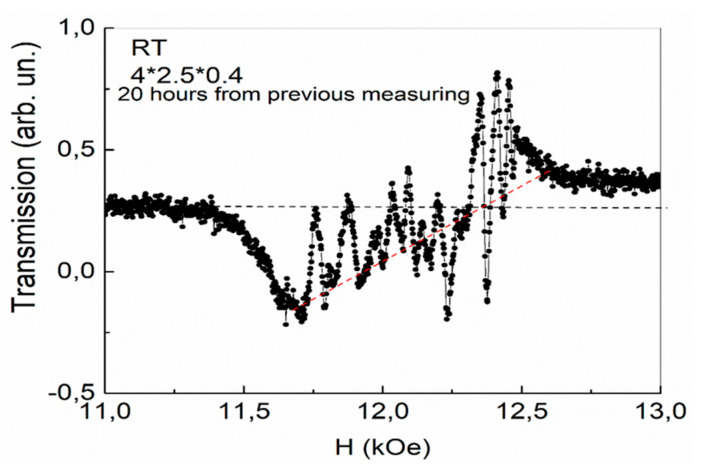
The phase separation regions in YFeG plate with dimensions of 4 × 3.5 × 0.4 mm after 20 h of spontaneous relaxation of the state shown in Figure 4.

**Figure 6 nanomaterials-13-02147-f006:**
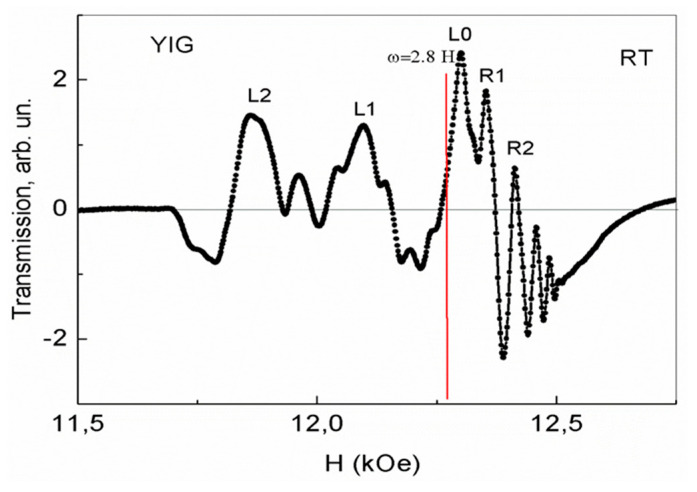
Equilibrium stable state of phase separation regions in YFeG plate with dimensions of 4 × 3.5 × 0.4 mm formed in the third cycle of magnetic field growth at a rate of 0.1 A/min.

**Figure 7 nanomaterials-13-02147-f007:**
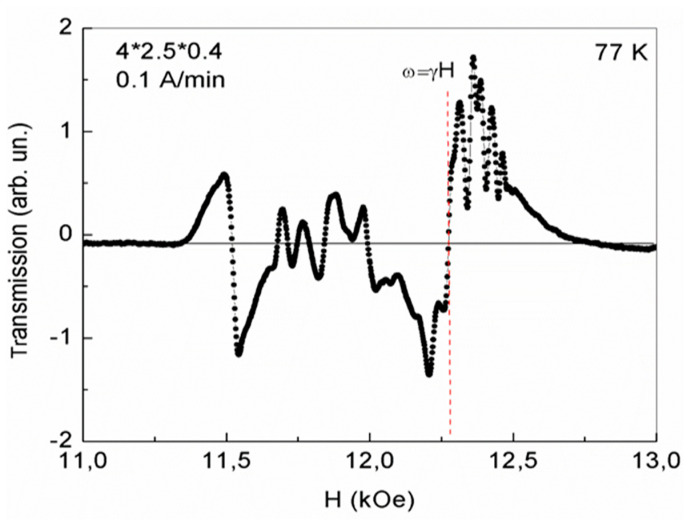
Excitation of de Haas–van Alphen oscillations in a 4 × 2.5 × 0.4 mm plate at temperature T= 77 K in 2D regions of phase separation with 2D electron gas in the third cycle of magnetic field growth at a rate of 0.1 A/min.

**Figure 8 nanomaterials-13-02147-f008:**
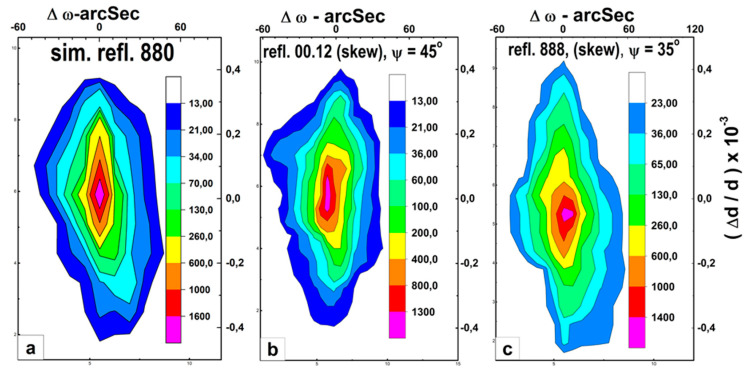
The measurement results presented in the form of reciprocal space maps (X-ray reciprocal space maps (RSMs)). Intensity distribution maps of the YFeG structure from the natural face of 110 symmetrical reflections: (**a**)—880, (**b**)—(skew. 00.12. ψ = 45°), (**c**)—skew. 888 ψ = 35°. The intensity distribution scale is shown on the right.

**Figure 9 nanomaterials-13-02147-f009:**
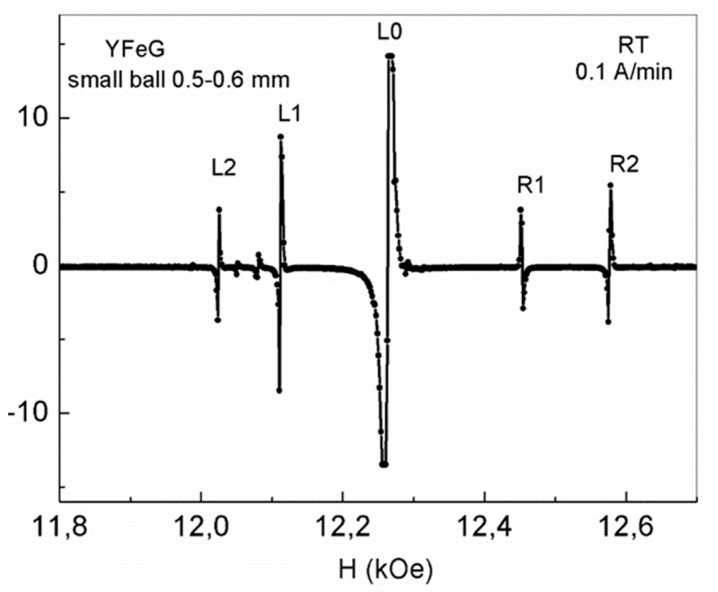
The 2D phase separation regions in the YFeG sphere with a diameter of 0.5–0.6 mm at room temperature at the frequency of 34.5 GHz after the third cycle of magnetic field growth at a rate of 0.1 A/min.

**Figure 10 nanomaterials-13-02147-f010:**
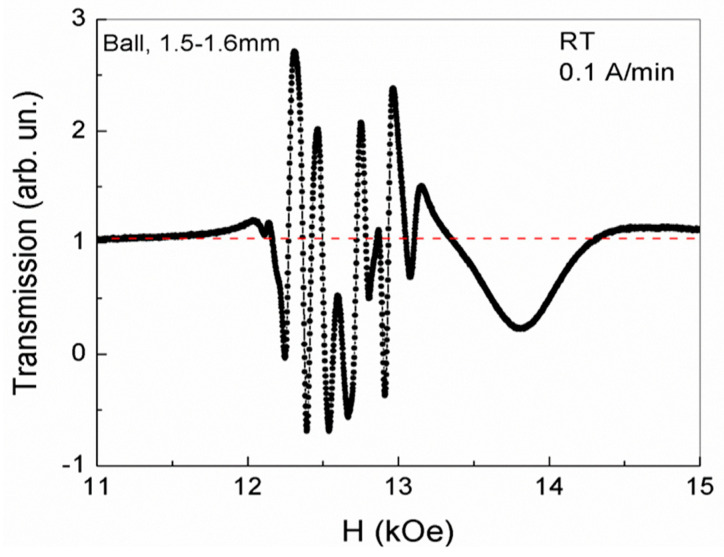
The phase separation regions at room temperature in the sphere with a diameter 1.5–1.6 mm during the third cycle of changing the magnetic field at a rate of 0.1 A/min.

## Data Availability

In the present study, references are provided to works in which one can find data confirming some of the specific results of this work. This is reported as the presentation of the main new results obtained in this paper. This is part of the main text. All new data is shared first time.

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
