# Peer review of "Landau Diamagnetism and de Haas–van Alphen Oscillations, Formed in Single Crystals of Y3Fe5O12, in Local Nanodimensional-Sized 2D Phase Separation Regions, Located inside Layered Domain Walls at Room Temperature and T = 77 K"

_nanomaterials, 2023, doi:10.3390/nano13142147_

Round 1

Reviewer 1 Report

This manuscript has reported the magnetic dynamics study of Y3Fe5O12 single crystals in the form of plates and spheres of various sizes. On the whole, the analysis and discussion of the mechanism is very thorough. I have a few questions for the author to improve the manuscript. How are Y3Fe5O12 single crystal samples of different sizes obtained? Should there be some specific physicochemical information about the Y3Fe5O12 single crystal sample? Some information about the characterization of Y3Fe5O12 single crystal samples should be added.

Author Response

Response to the comments of Reviewer No. 1.

Comment 1.

How samples of Y3Fe5O12 single crystals are obtained? How are samples of single crystals of different sizes and forms obtained? Some information about the characteristics of Y3Fe5O12 single crystals should be added.

Point 1.  Y3Fe5O12 crystals are grown by the solution-in-melt method. Vertical furnaces are used to grow crystals. The temperature regime in them is maintained with the required accuracy automatically by a software thermostat. In the zone of a moderate temperature gradient, conditions are created for the growth of large and homogeneous crystals.

       The technological process consists of determining the composition and preparation of the appropriate mixture, obtaining a homogeneous high-temperature solution, its slow cooling and mixing, and extraction of the formed crystals. To grow Y3Fe5O12 crystals, a charge is used (in molar %): 10% Y2O3; 20.4% Fe2O3; 36.8% PbO; 27.1% PbF2; 5.5% B2O3 [see “Single crystals of ferrites in radio electronics”, Yu.M. Yakovlev, S.Sh. Gendelev, Moscow, "Soviet Radio", 1975.]. In our studies, we used crystals grown by this method.

Point 2. These grown crystals had predominantly developed (110) faces, which are formed during the slow growth of crystals. Based on the equilibrium shape of the nucleus (110), the remaining main axes of the crystal were oriented in the directions (111) and (100), respectively. In these three directions, X-ray diffraction studies of Y3Fe5O12, presented in Fig. 8, were carried out.  They confirmed the presence of a Nano-sized superstructure of emerging 2D phase separation regions in the Y3Fe5O12 samples under study.

      Flat plates with polished surfaces for research were cut parallel to the developed planes of grown crystals. The spheres of various diameters were formed in special devises, in which these spheres were formed when compressed air was blown.

Reviewer 2 Report

B. Kh. Khannanov et al. present results of the magnetic dynamics (the microwave power absorptions at the fixed frequencies, during magnetic field sweeping) in samples of Y3Fe5O12 single crystals in the form of plates and spheres of various sizes, at frequencies exceeding 30 GHz, in magnetic fields up to 18 kOe, at room temperature and T=77 K. Overall, the idea received my attention and the methodology is technically sound. However, there are some specific issues the authors should address by making modifications before we can proceed and positive action can be taken.

  1. Author information: The name(s) of the author(s), The affiliation(s) of the author(s), i.e., institution, (department), city, (state), country.

  2. The decimal separators need to be consistent throughout the manuscript. The authors use point “.” in the main text while comma “,” in figures.

  3. Some sections can be divided into several subsections rather than presented in a huge section.

  4. The authors investigated the properties of magnetism. Have the authors noticed some papers documenting this point? i.e., [Magnetic behaviors of 3d transition metal-doped silicane: a first-principle study. J. Supercond. Nov. Magn. 2018, 31, 2789–2795, doi:10.1007/s10948-017-4532-4] and [Spin and valley half metal induced by staggered potential and magnetization in silicene. Chin. Phys. B 2014, 23, 017203]…

The English language requires improvements. Spelling and grammatical errors exist in the manuscript. i.e., the effective field Heff, which determined by…; in order to the phase separation regions to appear…; a complex pattern of 2D regions formed… We recommend you ask a native English speaker to edit the paper or use an independent professional editor.

Author Response

Response to the comments of Reviewer No. 2.

Comments.

Point 1. Information about the authors should be provided in more detail. Names of authors;organizations where the authors work;city;a country.

Response 1: Thank you for your comment - we have taken it into account.

Point 2. Decimal separators must be the same throughout the manuscript.Authors use a period in the body text, and a comma in numbers.We have taken this note into account.

Response 2: Thank you for your comment - we have taken it into account.

Point 3. Some sections can be divided into several subsections, rather than presented as a huge section.

Response 3: We took this remark into account and the most voluminous sections, which deal with similar, but still different phenomena, now have additional subheadings.

Point 4. The authors investigated the magnetic properties. Did they notice some manifestations of the properties characteristic of composites.

Response 4. In Y3Fe5O12, the properties, characteristic of composites, were not be observed.

Point 5. Editing the English language.

Response 5. We were edited English language.

         Thank you for the comments to Reviewer 2.

Reviewer 3 Report

The authors decribe the synthesis and properties of a garnet type structure Y3Fe5O12. There are two observations, aside from a thorough proofreading taking care of many annoying language issues:

- have the authors also run a basic XRD on their sample so as to quantify the phases (the ration garnet/perovskite?)

- On what basis have the Fe5+ - Fe+ pairs been utilized in the current work?

A more insightful comment on the potential applications of garnet type oxides should be presented as an outlook for those interested in the field.

There are several issues (rather minor) regarding the use of English language. See for instance in the introduction, where 's and plural "s"-ending are used interchangeable, brackets that open -line 180- without closing, etc. Most likely, a thorough proofreading will take care of this.

Author Response

Response to the comments of Reviewer No. 3.

Comment 1.  The authors describe the synthesis and properties of the Y3Fe5O12 garnet type structure.

Point 1.Have the authors performed basic X-ray diffraction analysis of your sample to quantify the phases (ratio garnet-perovskite phase).

Response 1: The Y3Fe5O12 crystals had predominantly developed (110) faces, which are formed during the slow growth of crystals. Based on the equilibrium shape of the nucleus (110), the remaining main axes of the crystal were oriented in the directions (111) and (100), respectively. In these three directions, X-ray diffraction studies of Y3Fe5O12 were carried out, presented in Fig. 8 in the reviewed article. They confirmed the presence of a nano-sized superstructure of emerging 2D phase separation regions in the Y3Fe5O12 samples under study. In this case, only the garnet phase was observed. No mixing of other phases was observed.

Comment 2. On what basis are Fe5+ - Fe1+ pairs used in Y3Fe5O12 in this work.

Response 2. In our previous work, referenced in [2] in this peer-reviewed paper, it is shown that the study of magnetic dynamics in Y3Fe5O12 revealed 2D phase separation regions, which turned out to be similar to phase separation regions in RMn2O5 multiferroics with charge ordering, WHICH ARE FORMED INSIDE THE DOMAIN WALLS. As shown in [7], referenced in this peer-reviewed paper, in the domain walls of magnetic crystals, flexi-elastic, magneto-electric inhomogeneity’s arise with an increased electron concentration, which changes the charge composition of ions in such walls. In [2] this case, such a change in the valences of 3d ions turns out to be energetically favorable, at which their charge ordering occurs. As a result, phase separation occurs in the domain walls of these crystals, similar to that formed in RMn2O5 multiferroics [2]. Since the phase separation regions in Y3Fe5O12 are formed in domain walls [2], it is advantageous to form charge ordering of Fe5+-Fe1+ ions in them. This was experimentally demonstrated in [2] and theoretically substantiated in [15].  

Comment 3. Provide an analysis of the applications of oxides such as garnet for those who are interested in this.

Response 3. When studying the ferromagnetic resonance (FMR) of Y3Fe5O12 (YFeG) single crystals, which have the form of spheres of a rather small diameter, at low microwave frequencies, in magnetic fields of the order of 3-4 kOe, it was found that YFeG behaves as a homogeneous, single-domain ferromagnetic with a narrow FMR line, which indicated on the low magnetic and dielectric losses [1]. For this reason, YFeG is the main material for applications in controlled microwave devices, operating at room temperature at sufficiently low microwave frequencies [1]. Our study at higher microwave frequencies, in stronger magnetic fields allowed us to obtain a new original result on the possibility of observing new effects in Y3Fe5O12 (Landau diamagnetism and de Haas van Alphen oscillations at room temperature), which are of scientific importance and, possibly, with practical applications will be discovered in time.

Round 2

Reviewer 2 Report

The authors have responded to the queries put to them, and they have also incorporated the necessary changes needed. The whole presentation is clearer. In light of the above, I would recommend the publication of this manuscript.